# High Concentrate Flavonoids Extract from Citrus Pomace Using Enzymatic and Deep Eutectic Solvents Extraction

**DOI:** 10.3390/foods11203205

**Published:** 2022-10-14

**Authors:** Johnatt Allan Rocha de Oliveira, Paula de Paula Menezes Barbosa, Gabriela Alves Macêdo

**Affiliations:** 1Nutrition College Federal University of Pará, University of Pará, Belem 66110-072, PA, Brazil; 2Department of Food and Nutrition, School of Food Engineering, University of Campinas, Campinas 13083-862, SP, Brazil

**Keywords:** deep euthetic solvents, phenolic compounds, flavonoids, extraction, betaine, cholin chloride

## Abstract

This paper evaluated methodologies for extracting phenolic compounds by DES (Deep eutectic solvents) associated with pectinlyase. Citrus pomace was characterized chemically, and seven DESs were formulated for extraction. Two groups of extractions were performed. Group 1 extractions were performed only with DESs, at 40 °C and 60 °C, with CPWP (Citrus pomace with pectin) and CPNP (Citrus pomace no pectin). In group 2, the DES was associated with pectinlyase and used only with CPWP at 60 °C in two ways of extraction: E1S (one-step extraction) and E2E (2-step extraction). The extracts were evaluated TPC (total phenolic compounds), individual phenolic compounds by HPLC, and antioxidant capacity by methodologies of DPPH and FRAP. The results of group 1 extractions for CPWP showed the highest phenolic compounds concentration (559.2 ± 2.79 mg/100 g DM) at 60 °C. Group 2 (E2S) showed high values of total phenolic compounds (615.63 ± 28.01 mg/100 g DM) and antioxidant activity (23,200 ± 721.69 µmol TE/g DM), with values higher than conventional extraction (545.96 ± 26.80 mg/100 g DM and 16,682.04 ± 2139 µmol TE/g DM). The study demonstrated the excellent extractive potential of DES for flavonoid extraction from citrus pomace. DES 1 and 5 by E2S showed the highest phenolic compounds and antioxidant capacity values, mainly when associated with pectinlyase.

## 1. Introduction

Brazil is the world’s largest orange producer with approximately 17.7 million tons of orange in the market year 2017/201, around 34% of the world’s production [1]. Within this productive chain, a large part of this material after the juice extraction, the main product in the chain, is released in the form of pomace, which corresponds to approximately 12.8 million tons. The orange pomace consists mainly of peel, bagasse, and seeds and is rich in phenolic compounds that can be used by the pharmaceutical industry as long as they are efficiently extracted [2].

The possibility of obtaining by-products with greater technological value has generated more and more studies for the use of pomace generated before and after pectin removal. One of the possible alternatives is to obtain bioactive compounds, aimed at the pharmaceutical and food industry [3].

Citrus by-products represent a cheap and abundant source of phenolic compounds, such as flavonoids and phenolic acids, since orange peels represent up to 65% of the fruit and have about 13.5 g of flavonoids/kg DM [4,5]. Important glycosylated flavanones and aglycones are found in this type of pomace; the first group is mainly represented by hesperidin and narirutin, and the second group by hesperetin and naringenin, respectively [6,7].

Based on the principles and metrics of green chemistry and sustainable development, it is essential to investigate green solvents with low toxicity and cost for the pharmaceutical and food industries. However, conventional organic solvents, for example, methanol, ethanol, ethyl acetate, and chloroform, are widely used in the extraction of bioactive compounds from natural plants [8]. The challenges of these volatile and hazardous organic solvents may lie in the environmental pollution produced and the unacceptable pomace of the solvent in the extracts.

One of the possible alternatives to organic solvents would be deep eutectic solvents (DES), such solvents were first introduced in 2003 by Abbott et al. [9] reporting mixtures of urea salts and quaternary ammonia, in particular the choline chloride (CC). The association between a halide salt or a hydrogen acceptor agent (HBA) with a hydrogen donor agent (HBD) is capable of forming a DES [10].

Despite some similarities between DES and ionic liquids, it is possible to notice some important advantages found for the first group with the second, of which it is possible to highlight the biodegradability and low toxicity, in addition to the low price and chemical inertness [10]. The most common DES are formed by choline chloride (ChCl) with some cheap and safe HBD, the most popular are urea, ethylene glycol, and glycerol or other alcohols, amino acids, carboxylic acids, and sugars are also commonly used [11].

The presence of functional groups such as hydroxyl, amide, and carboxylic groups are usually present in the composition of DES in large amounts, which favors the interaction between the H bond and the DES, favoring the extraction of phenolic compounds [12]. Zhang et al. [13] extracted catechin from green tea using DES. They found that although the extraction could be limited, the performance properties such as viscosity, surface tension, and polarity of DES allowed extractions at higher values than those performed with MeOH or water.

Qi et al. [14] evaluated the extraction of nine flavonoids from Equisetum palustre L. using nine different DES classified based on ChCl or betaine and observed that the best extraction yield of the target compounds was achieved when ethylene glycol was used as HBD, mainly with the combination of ethylene glycol with ChCl.

In our group, we have focused on demonstrating the potential of using orange pomace to offer extracts rich in phenolic compounds, in particular flavonoids, as well as by-products with high added value for the industry, obtained from different types of extraction or biotechnological treatments as those obtained via pressurized liquids [15]; fermented orange pomace using supercritical CO_2_ and cosolvent [16], tannase production using Brazilian citrus pomace [17], enzymatic biotransformation [18] among others.

The present work is justified by the opportunity to obtain more in-depth data on the application of different combinations of DES combined with enzymes since such liquids can produce extracts with a differentiated phenolic profile and are free of toxicity when compared with conventional extractions. These characteristics can be improved with the biotransformation promoted by the action of pectinlyase on the glycosidic fraction of the flavonoids, which can allow obtaining extracts rich in phenolic compounds and with high antioxidant capacity from a large volume of pomace citrus.

Thus, this work aimed to evaluate the extraction of flavonoids using different deep eutectic solvents (DES), associated in various combinations of extraction with pectinlyase, to promote the obtaining of extracts with varying profiles of flavonoids and antioxidant capacity from the pomace from the citrus industry.

## 2. Materials and Methods

### 2.1. Reagents and Samples

Hesperetin, naringenin and ellagic acid, 2,2-diphenyl-1-picrylhydrazyl (DPPH), TPTZ (2,4,6-tripyridyl-s-triazine), Trolox^®^, formic acid, choline chloride, betaine, and Pectinlyase were purchased from Sigma–Aldrich (St. Louis, MO, USA). Lactic acid, ethanol, citric acid, oxalic acid, maltose, and glycerol were purchased from Synth Co, and Citrus pomaces (CP) with pectin (CPWP) and no pectin (CPNP) were donated by CP Kelco industry (Limeira, SP, Brazil). The fresh samples were dried at 50 °C for 48 h and were milled. The powders used were that pass through a 9-mesh sieve and retained in a 10-mesh sieve and were stored at −4 °C for the experiments of the present work.

### 2.2. Determination of Chemical Composition of Pomace

All chemical analyzes of citrus pomaces were performed by the Association of Official Analytical Chemists methods [19]. Lipids content was determined by Bligh et al. [20]. All experiments were performed in triplicate and expressed as a percentage (%) based on dry matter. Total protein was determined by the Kjeldahl method [19], with a factor of 6.25 for protein conversion. The ash content was determined by incinerating the samples at 450 °C. The difference calculation was used to determine the content of (100 − (moisture + fat + protein + ash) [19].

### 2.3. DES Preparation

The DES used were prepared in the concentrations described in Table 1, using the heating-stirring method proposed by Abbot et al. [9] with slight modifications. The DES was prepared by mixing the individual components in a Becker of 500 mL, under heating in a water bath at 80 °C, with agitation performed with the aid of turrax. After the mixtures were prepared, they were used as extractors.

### 2.4. Extraction Procedures

The extractions were separated into two groups: Group 1 and Group 2 and the procedures for each one are shown in Figure 1. In the group 1 extractions, two citrus pomaces were used: CPWP and CPNP with temperatures of 40 and 60 °C. Ethanol:water (1:1; *v*/*v*) was also used as extraction solvents under the same conditions, for comparison purposes. In group 2 extractions, only CPWP was used at a temperature of 60 °C. In this group, the extraction occurred with the DES developed in association with the pectinlyase Novozym 33095^®^ (Novozymes, (Araucária, Brazil)) the enzymatic load was 40 μL/100 mL.

### 2.5. Extracts Characterization

#### 2.5.1. Total Phenolic Compounds (TPC)

The Folin-Ciocalteu method was used according to Singleton et al. [21], and the assay was carried out on a NovoStar Microplate reader (BMG LABTECH, Offenburg, Germany) with a filter of 725 nm. A standard calibration curve was constructed with gallic acid (Sigma-Aldrich (St. Louis, MO, USA)) in a concentration range of 25–200 µg/mL. The analyzes were performed in triplicate and the results were expressed in mg of gallic acid equivalents (GAE)/mL or mg of extract.

#### 2.5.2. Determination of Main Phenolic Compounds by High-Performance Liquid Chromatography (HPLC)

The phenolic compounds were quantified according to the methodology described by Madeira et al. [22]. To identify and quantify the main phenolic compounds present in the extracts, a Dionex UltiMate 3000 chromatograph (Dreieich, Germany) was used, using a C-18 Acclaim 120 column (Dionex, 3 µm, 4.6 × 150 mm) at a temperature of 30 °C. The equipment allowed to quantify the phenolic compounds: Naringin, Narirutin, Hesperidin, Ellagic Acid, Naringenin, Hesperitin, Diosmetin, Tangerine, and Gallic Acid. For the experiment, it was necessary to use the following mobile phases: A-H2O (0.1% formic acid) and B-Methanol (0.1% formic acid) with a flow rate of 0.6 mL.min-1. A diode array detector (DAD-3000) allowed detection at 280 nm. Thus, the identification of flavonoids was based on the comparison of retention times and UV-VIS spectra of the standards of such compounds evaluated.

#### 2.5.3. DPPH and FRAP Radical-Scavenging Activity

The DPPH method (2,2-diphenyl-1-picrylhydrazyl stable free radical) was used to measure the antioxidant capacity of the extracts and followed the methodology described by Macedo et al. [23]. The reaction mixture was read in microplates of 96 wells using the NovoStarMicroplate reader (BMG LABTECH, Offenburg, Germany) at an absorbance of 520 nm. The reaction mixture used for reading in the equipment followed the sequence: 50 µL of the test sample was mixed with 150 µL of DPPH prepared at 0.2 mM in methanol.

The antioxidant activity was measured by the discoloration of the DPPH reagent, after 90 min of reaction. The results were calculated from the equation resulting from the linear regression after plotting the solutions of known Trolox concentration. The results were expressed in μmol equivalent of Trolox (TE)/mL of dry matter and all measurements were performed in triplicate.

The determination of antioxidant activity by the FRAP method. The FRAP reagent was prepared by mixing 25 mL of acetate buffer (0.3M, pH 3.6) (Riedel-de Haen, Seelze, Germany) with 25 mL 10 mM TPTZ (2,4,6-tripyridyl-s-triazine; Fluka Chemicals, Buchs, Switzerland) 25 mL of 20 mM FeCL_3_ solution. The methodology was carried out according to Benzie et al. [24]. In Eppendorf, 30 μL of standard or white samples, 90 μL of distilled water, and 900 μL of the FRAP reagent were pipetted. The mixture was vortexed and 200 μL aliquots were transferred to a 96-well transparent microplate. Absorbance was measured in a microplate reader (FLUOstar OPTIMA; BMG Labtech, Offenburg, Germany) at 595 nm and 37 °C for 10 min. The standard curve was previously prepared with Trolox^®^ solutions in concentrations between 15 and 1500 μmol/mL. The results were expressed in μmol equivalent of Trolox (TE)/mL of dry matter and the analyzes were performed in triplicate.

### 2.6. Statistical Analysis

For statistical evaluation of the results, the mean value and standard deviation of the triplicates were calculated. In addition, a comparison was performed between the samples using the *t*-test (*p* ≤ 0.05) using the Minitab 16.1.1 statistical program.

## 3. Results and Discussion

### 3.1. Chemical Composition

Table 2 shows the results of the chemical characterization of the citrus pomace used.

When comparing the chemical characterization of citrus pomace with and without pectin, it was possible to observe that the pomace with pectin presented higher values of ash and proteins than the pomace without pectin, which in turn, showed higher values of moisture, lipids, and carbohydrates, that the material with pectin. According to Barbosa et al. [3], the ash content of the orange pomace with pectin is up to 4× higher than that of pomace without pectin. This information is similar to what was verified in the present work since the ash content in the CPWP was 3.18 times higher than in the CPNP.

Except for carbohydrate content, all other components of chemical characterization were statistically different at 0.05% significance. Both the evaluated materials verified low values of proteins and high carbohydrate levels. According to Aravantinos-Zafiris et al. [25], both pomaces can be considered valuable sources of fiber for food or food ingredients. Thus, associating this aspect with the possibility of interesting values of bioactive antioxidant compounds extracted through innovative and economically viable extraction methods can further enhance these pomaces.

### 3.2. Results Obtained for Non-Enzyme Assisted Phenolic Extraction in Deep Eutectic Solvents-Group 1

#### 3.2.1. Total Phenolic Compounds (TPC)

Table 3 shows the results of the total phenolic compounds obtained for the extraction performed with the eutectic mixtures selected for pomace with and without pectin, respectively.

Ethanol:water mixture extraction was used as the reference solvent, as hydroalcoholic mixtures are commonly used as extraction solvents to recover phenolic compounds [26]. It is possible to verify that the values of TPC (total phenolics compounds) were higher for CPWP, at both temperatures used and for all applied DES, which can be explained by the fact that in the industrial pectin extraction process, the citrus pomace is washed consecutively with strong acid solutions such as hydrochloric and sulfuric, among others [27], which generates a removal of the phenolic compounds and other compounds of interest present.

The concentration of TPC varied between 159.87–545.96 mg/100 g DM for extraction with ethanol:water, while the extraction with the use of DES varied between 86.39–559.20 mg/100 g DM. The temperature of 60 °C allowed greater extraction of TPC, for both pomaces, with a significant difference (*p* < 0.05) observed for most of the DES assessed. The increase in temperature can improve the solubility of phenolic compounds and their rate of diffusion in the solvent, thus increasing the rate of mass transfer [28,29]. Temperature is the second most crucial variable in the phenolic extraction process with the use of DES, after the water content, since both interfere positively with the viscosity of the DES at the time of extraction, improving the mass transfer of bioactive molecules from natural matrices [30,31]. DES 1 released higher levels of TPC from CPWP and CPNP at both temperatures applied, followed by DES 5, which has glycerol in its formulation and is very important for these results, as this component is usually used in the formulation of several foods without any danger for their consumption in pre-determined quantities. Ozturk et al. [32] evaluated the TPC extraction from citrus pomace. They observed that the glycerol chloride mixture (molar ratio 1:2) obtained the second-highest concentration of TPC with 334 mg GAE/100 g, surpassing even the ethanolic mixture and similar to our work, proving the excellent extraction potential of this mixture. It is possible to verify that some of the evaluated DES allowed better extractions of TPC than the conventional extraction performed with water and methanol 1:1, such as DES 1, 4, 5, and 6, for CPWP, at a temperature of 40 °C and DES 1 and 5 for the same pomace at a temperature of 60 °C. For CPNP, this behavior was observed for DES 1, 4, 5, and 6 at 40 °C and DES 1 and 5 at 60 °C. Concerning betaine-based DES, organic acid and polyalcohol were selected as HBDs. Betaine-based DESs, in association with lactic acid, showed extraction with greater efficiency than that of ChCl-based DES, for most of the tested DES, at both temperatures and evaluated pomace. For betaine-glycerol DES, it was also possible to observe good extractions of TPC compared to other DES based on ChCl. The good results obtained by the extractions performed with the use of betaine, mainly in association with lactic acid, are related to the fact that it has reversibly protonated functionality when reducing the pH, which would allow aspects such as: allowing the formation of micelle-like clusters, through self-assembly; act as hydrophilic and hydrophobic nano-containers, in addition to the fact that they are subject to surface loading and size variations according to pH [33].

The highest value of TPC released was 559.2 ± 2.79 mg/100 g DM, obtained with DES 1, at a temperature of 60 °C, from the CPWP, which represents an increase of 2.42% compared to the highest TPC value obtained in conventional extraction. While the lowest TPC value was found for DES 3 and 7 from the CPNP, with no statistical difference between the two extractions. Barbosa et al. [3] found for the same materials used in the present work, the maximum values of 386 and 170 mg/100 g DM, for CPWP and CPNP respectively, with the conventional extraction performed with water:ethanol-(1:1). Dai et al. [34] used a variety of natural DESs to extract polar and less polar phenolic metabolites from safflower and observed greater extraction performance of several DES compared to conventional solvents, which demonstrates the potential of this process.

#### 3.2.2. Phenolic Compounds Profile

Phenolic compounds are complex molecules, and their extraction from a solid matrix requires compatible solvents. Hence, it is common for different compounds to be more or less extracted with the use of different solvents. The quantification of the main phenolic compounds of each extract is shown in Figure 2. It was found that the two main phenolic compounds extracted with both pomaces used for all DES evaluated were hesperidin and narirutin, mainly in extracts obtained with the CPWP, which coincides with the results obtained by Khan et al. [35].

When we compare the phenolic compounds obtained from both materials, we find that Hesperidin was the one identified in greater quantity in CPWP, with a significant decrease in its quantity and that of narirutin compared to CPWP extracts. This difference is caused by the process of consecutive washes with acidic solutions to extract pectin, which may also explain the absence and decrease in the quantity of the other flavonoids analyzed.

Gallic acid is considered one of the phenolic compounds of most significant industrial interest, so it is necessary to report that DES 1 (Betaine:lactic acid; 1:1), together with DES 2 and 3, were the only ones that allowed its extraction from both pomaces, being possible that there is an affinity of these DES such a compound. For CPWP, none of the DES allowed extracts with gallic acid content in concentrations greater than 34 mg/100 DM, as observed for CPWP, probably due to their removal during pectin production.

Betaine was chosen, as an alternative to choline chloride, in the production of DES since it is a plant-based compound, cheaper and less toxic than choline chloride, biodegradable, and has an extensive potential application [36,37,38]. Krisanti et al. [39] observed higher levels of phenolic compounds extracted with betaine DES from ground coffee, the values higher than the results obtained with conventional extraction performed with ethanol. There are still few results in the literature that justify the beneficial effects in the extraction with the betaine:lactic acid association, which was observed is that the interaction between these molecules together with the addition of water, has a strong extraction power of compounds of interest, such as phenolic compounds [40,41]. According to Yazici et al. [42], the extraction of phenolic compounds from Capparis Ovata var canescens fruit reached the highest values when performed with DES composed of lactic acid combined with ChCl, with a liquid/solid ratio of 20:1 (mL/g).

#### 3.2.3. Antioxidant Capacity

Table 4 and Table 5 show the results of FRAP and DPPH found for extracts obtained in group 1 at different temperatures (60 °C and 40 °C) for CPWP and CPNP, respectively.

It was observed in Table 4 that the CPWP showed higher values of antioxidant activity in most of the DES used, which is explained by the more significant number of phenolic compounds present in the material, which did not go through the pectin extraction process. DES 1 was the one that allowed the highest values of FRAP and DPPH, which were 59,100.0 ± 2243.13 and 9661.12 ± 168.58 FRAP (µmol TE/g DM), respectively, both at a temperature of 60 °C, not observing statistical difference for the values of DPPH obtained for the CPWP in the different evaluated temperatures, which would allow to carry the process through in temperatures of 40 °C, without prejudice of the antioxidant capacity for this eutectic combination.

In Table 5 for CPNP, the highest value of antioxidant activity found was 23,100.0 ± 3651.93 and 3131.40 ± 98.80 for FRAP and DPPH, respectively, with no statistical difference at 95% confidence, between the FRAP values obtained at temperatures 40 °C and 60 °C, for the extracts obtained with DES 1, which was not observed for the values of DPPH in the same condition. For both CPWP and CPNP, a decreasing scale of values was found according to the DES used, with a decreasing sequence being observed for the group of extracts analyzed for FRAP and DPPH, so the sequences for CPWP/FRAP and DPPH/at 40 °C and 60 °C-DES1 > DES5 > DES4 > DES 6 > DES 2 > DES7 > DES3. For CPNP, the following sequence was observed for FRAP/at 60 °C: DES1 > DES4 > DES5 > DES6 > DES2 > DES7 > DES3, while for the other values of FRAP at 40 °C and DPPH at 40 °C and 60 °C the sequences were the same that observed for the CPWP.

It was found that most of the extractions performed at 60 °C showed higher values of antioxidant activity for both methods of analysis. Only the extraction performed with DES1 obtained higher values than the extractions performed with ethanol:water (1:1), for FRAP and DPPH, with DPPH values at 60 °C, found for CPWP and CPNP, 96,551.04 and 2943.01 µmol TE/g DM, respectively, which were higher than those observed by Barbosa et al. [3], which were 11,035 and 2571 µmol TE/g DM, for the same material.

### 3.3. Results for Phenolic Enzymatic Assisted Extraction in Deep Eutectic Solvents-Group 2

#### 3.3.1. Total Phenolic Compounds (TPC)

For group 2 extractions, only the CPWP was used at a temperature of 60 °C due to the best results observed in group 1 extraction. Table 6 shows the results of the total phenolic compounds found for the extractions performed with the eutectic mixtures associated with the pectinlyase enzyme.

The modification in the extraction methodology around the time of adding the enzyme did not promote a significant difference for most extraction pairs with DES 1, 3, and 5. However, for most extractions, the modification in the extraction sequence led to values of higher phenolic compounds, observed for DES 2, 4, 6, and 7. The highest values of phenolic compounds were obtained for DES 1 and 5, with the highest value obtained for the extraction performed with DES 1-E2S, with the value of 615.63 mg/100 g DM, without a statistical difference (*p* < 0.05) in comparison to its pair, in 1-step extraction, which reached the value of 604.53 mg/100 g DM, it is essential to highlight that both values mentioned are superior and with a statistical difference when compared to the value obtained with the conventional extraction, which reached the value of 545.96 mg/100 g DM.

The worst extraction value in group 2 was found for DES 7 and DES 3 by E1S. Although much literature on the extraction of phenolic compounds reports the positive effect of the pH reduction for the extractive effect [29], this behavior was not verified since both DES are constituted by acids and obtained reduced values. However, when we observe the same DES in E2S, it is clear that the extraction separation in two steps improved the extraction values since the pectinlyase acted separately at its optimal pH, without the interference of the eutectic mixture in the extractive medium as occurred in the extractions by E1S.

#### 3.3.2. Individual Phenolic Compounds–Phenolic Compounds Profile

Table 7 and Table 8 show the TP values for group 2 extractions. For important phenolic compounds, such as gallic acid, conventional extraction obtained the highest value of 34.60 mg/100 g DM, followed by the value of DES 2-E2S (6.03 mg/100 g DM) and soon afterward by DES 5-E1S (6.02 mg/100 g DM), with the appearance of gallic acid being observed more frequently in extracts obtained with the associated extraction, than that performed only with DES, in group 1 extractions.

As was seen for group 1 extractions, the phenolic compounds obtained in higher concentrations in group 2 were hesperidin and narirutin, both glycosylated flavonoids. Although the flavonoids naringenin and hesperitin for groups 1 and 2 demonstrated differences, with an increase in the concentrations of both flavonoids in the aglycone form for E2S extraction. Thus, the concentrations of naringenin and hesperetin in extraction group 1 (CPWP at 60 °C—not shown in the table) ranged from 3.36–4.48 and 9.36–13.17 mg/100 g DM, respectively. While in group 2 by E2S, the concentrations of the same compounds varied between 22.35–82.26 and 20.31–83.55 mg/100 g DM for naringenin and hesperitin, respectively, indicating the effect of pectinlyase inserted in all DES applied by E2S.

The same behavior was also observed in group 2 for extractions using the E1S method but in a smaller proportion. This was caused by the fact that in the E2S, the enzyme remains longer in its optimal condition when it is acting only in the presence of the buffer, without the eutectic mixture, which does not happen in the E1S extraction, where the entire activity of the enzyme occurs in the presence of DES, showing that in this sense, the presence of DES negatively interfered in the action of pectinlyase and the biotransformation of the mentioned flavonoids. According to Park et al. [43], pectinlyase was able to promote an increase in the concentrations of naringenin and hesperitin in extracts obtained from the citrus peel, with a positive impact on the antioxidant activity of the extracts, similar to what was observed in this work.

In Figure 3, Figure 4 and Figure 5, it is possible to observe the chromatograms obtained for the extraction carried by solvent (water:ethanol, 1:1) and the extractions carried out by DES3 for the extraction carried by one step (E1S) and for the extraction performed by two steps (E2S).

In Figure 3, Figure 4 and Figure 5, an improvement in the peaks was observed, mainly of naringenin and hesperitin, for extraction by E2S, which demonstrates the effect of the enzyme on the release of such compounds, which is not observed for the peaks obtained from extraction with solvents or extraction E1S, possibly by reducing the activity of the enzyme, when immersed in DES3. The extraction with DES3 in two-step allowed the enzyme has a good action since it was used separately in its use buffer and later with DES 3, which was confirmed by the appearance of the flavone aglycone diosmetin, not observed in Figure 1 and Figure 2, possibly due to the biotransforming effect of pectinlyase from the removal of the aglycone group from the flavonoids present in the analyzed extract. Each DES used allowed different profiles of extracted compounds. For DES3, it was possible to demonstrate the effect of enzyme when in the presence of the buffer due to the two-step extraction (E2S), which provided the appearance of diosmetin in this extraction and not appearance for the solvent extraction and one-step extraction (E1S), which demonstrates that the use of pectinlyase can be limited when used directly in the presence of DES, and makes the two-step extraction more recommended if the interest is also in the excellent functioning of pectinlyase and its biotransformation effects.

#### 3.3.3. Antioxidant Activity

Table 9 shows the values of the antioxidant activities obtained for the extracts resulting from group 2 extractions, performed in association with the enzyme pectinlyase.

The highest values of FRAP and DPPH were 23,200 ± 721.69 and 10,545.38 ± 75.47 µmol TE/g DM, respectively, and were observed for DES 1, followed by DES5 values, mainly in E2S. This result is due to the higher concentration of phenolic compounds obtained for such extracts. Besides, it was found that most of the results of antioxidant capacities were also more significant in the extraction in E2S, as this allowed the greater release of non-glycosylated flavonoids, such as hesperitin and naringenin, in addition to detectable concentrations of gallic acid for most of the DES used, different from what happened at E1S. Thus, it is essential to mention that the high values of antioxidant capacity obtained for E2S are the result of more efficient biotransformation compared to E1S due to the effect of the pectinlyase action, which was verified in item 3.3.2 since no statistical difference was found for the TPC values obtained in the extractions E1S and E2S para o DES5. This positive effect on the antioxidant activity value for DES5 ([Ch]Cl: Gly) was also observed by Ozturk et al. [32], which obtained the second-highest value of antioxidant capacity by DPPH with this mixture from citrus pomace. According to Moore et al. [44], solvent characteristics, such as polarity, pH, hydrogen capacity, or the ability to supply hydrogen atoms to free radicals are factors that affect the free radical scavenging reaction mechanism, which significantly affects the overall kinetics test and the results of antioxidant capacity.

A statistically significant difference was found for most of the results obtained for FRAP in E1S and E2S. However, this behavior was not repeated for the DPPH results, with no statistical difference found for many of the extractions. The extractions of group 2 allowed many values of antioxidant activity higher than those with conventional solvents, demonstrating the excellent potential for applying the mixtures developed in the different forms of extractions evaluated. That technology demonstrates even more potential since many of the mixtures used are composed of components for food use and low toxicity, which can be applied for compounds of interest extraction, such as flavonoids, allowing obtaining safe extracts with high antioxidant capacity.

## 4. Conclusions

It was possible to conclude that the extraction with Eutectic Solvents of group 1 extracted high values of phenolic compounds, mainly for DES1, and that the CPWP material released higher values of extracted flavonoids, hesperidin and narirutin, removed in more significant amounts at the temperature of 60 °C. The CPWP material was selected for the group 2 extractions, assisted by enzymes and associated with DES. For group 2 extractions, DESs 1 and 5 showed the highest levels of total phenolic compounds extracted, more elevated than conventional extraction. The addition of pectinlyase increased the concentration of aglycone flavonoids in group 2, mainly for the E2S extraction, and promoted a significant increase in the concentration of naringenin and hesperitin, with increased antioxidant capacities, in both methods, DPPH and FRAP. The DES composed of betaine/lactic acid and choline chloride/glycerol were the most effective in extracting total phenolic compounds, with values higher than the extraction with conventional solvent. Thus, the proposed methodologies showed promising results and can be used as ecologically correct and safer alternatives than traditional extractions. The association of pectinlyase with DES, through the extraction in the E2S configuration, proved to be an efficient strategy for the extraction of phenolic compounds and allowed the obtainment of extracts rich in flavonoids and bioactive compounds of high industrial value, which create an alternative for the use of the large volume of citrus bagasse generated by the processing industries.

## Figures and Tables

**Figure 1 foods-11-03205-f001:**
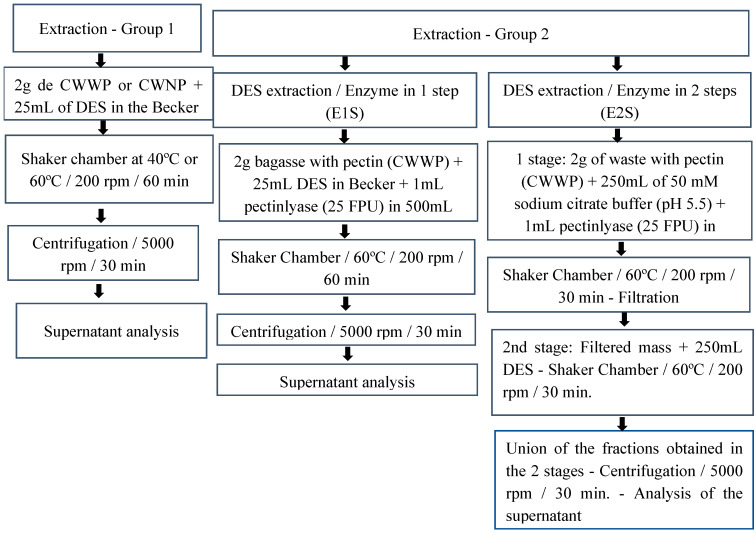
Scheme of extractions performed.

**Figure 2 foods-11-03205-f002:**
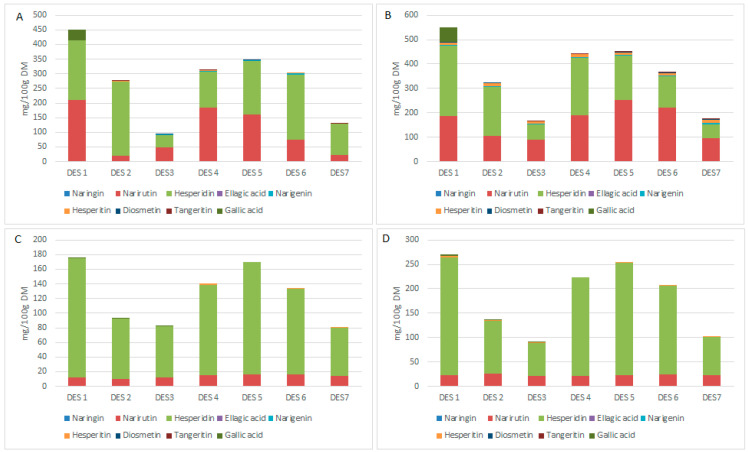
Individual phenolic compounds extracted from CWWP at 40 and 60 °C (**A**,**B**); Individual phenolic compounds extracted from CWNP at 40 and 60 °C (**C**,**D**).

**Figure 3 foods-11-03205-f003:**
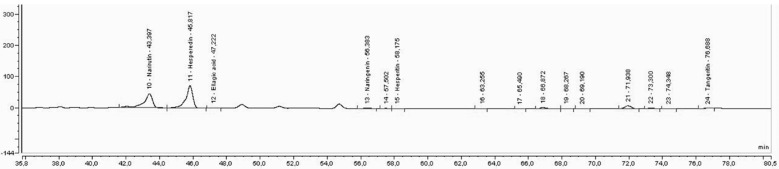
Chromatogram of the extraction performed by solvent.

**Figure 4 foods-11-03205-f004:**
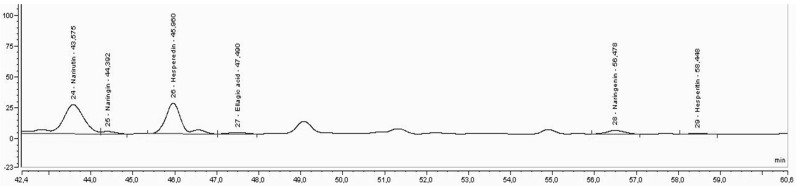
Chromatogram of the extraction performed with DES3 by one step.

**Figure 5 foods-11-03205-f005:**
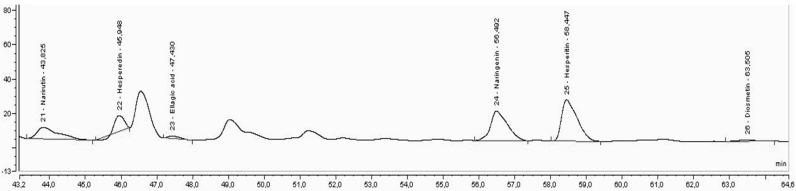
Chromatography of the extraction performed with DES3 in two steps.

**Table 1 foods-11-03205-t001:** Composition of DES.

Abbreviation	Components		Mol Ratio	pH
	Component 1	Component 2		
DES 1	Betaine	Lactic acid	1:1	3.5
DES 2	Choline chloride	Citric acid	2:1	3.8
DES 3	Choline chloride	Lactic acid	1:2	3.6
DES 4	Choline chloride	Maltose	1:2	5.8
DES 5	Choline chloride	Glycerol	1:2	5.9
DES 6	Betaine	Glycerol	1:2	5.7
DES 7	Choline chloride	Oxalic acid	1:2	3.8

**Table 2 foods-11-03205-t002:** Chemical characterization of citrus pomace with and without pectin.

Component (g/100)	Citrus Pomace with Pectin (CPWP)	Citrus Pomace No Pectin (CPNP)
Moisture	7.02 ± 0.09 a	7.51 ± 0.08 b
Ashes	3.12 ± 0.02 a	0.98 ± 0.05 b
Proteins	7.11 ± 0.10 a	6.65 ± 0.30 b
Lipids	3.98 ± 0.20 a	4.35 ± 0.20 a
Carbohydrates	78.77 ± 0.08 a	80.51 ± 0.07 a
Total	100	100

CPWP Citrus pomace with pectin, CPNP Citrus pomace no pectin; a, b Values (mean ± SD) within each analyte with different letters significantly differ (*p* < 0.05), tested using *t*-test.

**Table 3 foods-11-03205-t003:** Phenolic compounds for citrus pomace with pectin.

DES	Total Phenolic Compounds (mg/100 g DM)–CPWP		Total Phenolic Compounds (mg/100 g DM)–CPNP	
	Extraction at 40 °C	Extraction at 60 °C	Extraction at 40 °C	Extraction at 60 °C
Water:ethanol (1:1)	381.34 ± 9.26 g	545.96 ± 4.66 f	159.87 ± 3.40 h	259.45 ± 3.45 h
DES 1	463.34 ± 7.88 b	559.2 ± 2.79 a	266.91 ± 4.14 a	273.30 ± 4.52 a
DES 2	333.44 ± 5.47 d	335.32 ± 9.36 d	99.32 ± 0.72 fg	143.92 ± 5.96 e
DES 3	99.50 ± 7.24 f	172.64 ± 6.52 e	86.39 ± 1.40 g	101.74 ± 6.86 fg
DES 4	395.23 ± 9.65 c	470.75 ± 7.24 b	202.98 ± 3.88 cd	226.40 ± 5.13 b
DES 5	402.89 ± 1.71 c	465.84 ± 6.17 b	219.73 ± 0.79 bc	271.26 ± 3.62 a
DES 6	388.15 ± 9.86 c	405.71 ± 2.96 c	199.78 ± 0.49 d	211.04 ± 9.48 bcd
DES 7	167.65 ± 2.54 e	181.97 ± 5.13 e	87.66 ± 2.09 g	110.31 ± 5.81 f

Results are presented as the mean (*n* = 3) ± SD, and those with different letters are significantly different, with *p* < 0.05.

**Table 4 foods-11-03205-t004:** Results of antioxidant capacity for group 1 extractions from CPWP.

DES or Extractive Mixture	FRAP (µmol TE/ g DM)		DPPH (µmol TE/ g DM)	
	Extraction at 40 °C	Extraction at 60 °C	Extraction at 40 °C	Extraction at 60 °C
Ethanol:water (1:1)	52,113.15 ± 298.33 c	53,985.00 ± 643.13 bc	8710.03 ± 122.21 b	9655.04 ± 174.45 a
DES 1	55,770.8 ± 500.99 b	59,100.0 ± 2243.13 a	8845.07 ± 143.46 b	9661.12 ± 168.58 ab
DES 2	38,129.2 ± 73.06 d	36,662.5 ± 1110.27 de	2485.10 ± 466.87 f	2456.87 ± 183.81 f
DES 3	33,541.7 ± 187.50 fg	31,930.0 ± 325.36 g	2061.70 ± 58.65 f	2256.49 ± 212.20 f
DES 4	38,804.2 ± 338.19 d	38,904.2 ± 171.85 d	4204.80 ± 159.80 e	8164.04 ± 191.58 bc
DES 5	52,029.2 ± 318.93 c	53,833.3 ± 732.33 bc	7108.45 ± 157.85 d	8499.13 ± 234.44 b
DES 6	38,295.8 ± 1789.04 d	37,216.7 ± 752.91 de	4181.43 ± 159.69 e	7411.06 ± 292.27 cd
DES 7	35,154.2 ± 279.23 ef	34,537.5 ± 473.74 ef	1253.26 ± 126.75 g	2443.92 ± 156.60 f

Results are presented as the mean (*n* = 3) ± SD, and those with different letters are significantly different, with *p* < 0.05.

**Table 5 foods-11-03205-t005:** Results of antioxidant capacity for group 1 extractions from CPNP.

DES or Extractive Mixture	FRAP (µmol TE/ g DM)		DPPH (µmol TE/ g DM)	
	Extraction at 40 °C	Extraction at 60 °C	Extraction at 40 °C	Extraction at 60 °C
Ethanol:water (1:1)	19,103 ± 145.12 abc	22,912.05 ± 774,90 a	2301.01 ± 202.30 d	2943.01 ± 154.80 b
DES 1	20,773.42 ± 231.27 ab	23,229.2 ± 774.90 a	2215.20 ± 162.30 d	3131.40 ± 98.80 b
DES2	17,808.94 ± 325.81 de	22,062.5 ± 493.76 ab	1133.60 ± 33.30 e	2118.10 ± 109.40 d
DES 3	15,696.95 ± 391.34 f	18,370.8 ± 170.63 de	842.40 ± 31.10 fg	1164.60 ± 148.20 f
DES 4	19,183.09 ± 510.92 cd	22,520.8 ± 202.07 ab	1334.10 ± 56.80 ef	2603.40 ± 36.50 c
DES 5	19,494.43 ± 268.95 bc	23,100 ± 3651.93 a	1371.2 ± 152.60 ef	2756.0 ± 200.00 a
DES 6	18,335.67 ± 243,91 de	22,333.3 ± 966.42 ab	1215.90 ± 35.10 ef	2600.80 ± 111.90 c
DES 7	16,693.02 ± 338.89 ef	20,900 ± 817.04 abc	1066.70 ± 58.00 f	2082 ± 57.20 d

Results are presented as the mean (*n* = 3) ± SD, and those with different letters are significantly different, with *p* < 0.05.

**Table 6 foods-11-03205-t006:** Total phenolic compounds (TPC) for extraction supplemented with pectinlyase-Group 2.

DES or Extractive Mixture	Total Phenolic Compounds (mg/100 g DM)	
	E1S	E2S
Water:ethanol (1:1)	545.96 ± 26.80 b	-
DES 1	604.53 ± 14.13 a	615.63 ± 28.01 a
DES 2	270.95 ± 13.08 fg	323.14 ± 20.60 e
DES 3	218.21 ± 11.64 h	234.69 ± 14.36 gh
DES 4	460.60 ± 14.86 d	545.20 ± 28.20 bc
DES 5	560.58 ± 26.80 b	544.30 ± 17.61 b
DES 6	348.41 ± 11.88 e	500.16 ± 10.60 cd
DES 7	96.42 ± 5.03 i	258.19 ± 9.37 ef

Results are presented as the mean (*n* = 3) ± SD, and those with different letters are significantly different, with *p* < 0.05. E1S–extraction one step; E2E–Extraction 2 steps.

**Table 7 foods-11-03205-t007:** Individual phenolic compounds obtained for group 2 extractions by One extraction step–E1S.

Phenolic Compounds (mg/100 g DM)	Water:Ethanol (1:1)	DES1	DES2	DES3	DES4	DES5	DES6	DES7
Naringin	nd	5.76 ± 0.73 a	nd	3.36 ± 0.04 b	5.30 ± 0.27 a	nd	3.25 ± 0.01 b	nd
Narirutin	127.91 ± 23.42 a	203.79 ± 12.46 b	32.45 ± 1.89 bc	47.17 ± 0.32 de	279.93 ± 3.11 c	188.24 ± 2.4 ef	153.66 ± 4.60 d	24.16 ± 1.01 f
Hesperidin	252.21 ± 8.29 a	184.33 ± 7.95 b	45.04 ± 0.57 hi	48.39 ± 1.08 hi	83.70 ± 6.40 g	214.56 ± 9.49 ef	77.88 ± 0.04 i	15.80 ± 0.14 j
Ellagic acid	nd	2.97 ± 1.59 def	4.47 ± 0.41 d	7.70 ± 0.07 bc	2.53 ± 0.23 ef	nd	9.00 ± 0.81 ab	4.10 ± 0.18 de
Naringenin	2.37 ± 0.04 g	nd	24.31 ± 2.14 cde	24.64 ± 0.15 cde	4.89 ± 0.14 g	nd	22.59 ± 0.08 def	22.35 ± 0.40 ef
Hesperitin	1.34 ± 0.04 h	nd	18.11 ± 0.18 g	16.50 ± 0.10 g	1.46 ± 0.04 h	nd	22.77 ± 0.15 de	20.31 ± 0.42 ef
Diosmetin	0.85 ± 0.03 d	nd	3.39 ± 0.39 b	nd	nd	nd	3.36 ± 0.04 b	3.03 ± 0.03 bc
Tangeritin	0.99 ± 0.07 f	1.73 ± 0.00 c	1.14 ± 0.02 e	1.18 ± 0.02 e	1.44 ± 0.01 d	1.78 ± 0.00 bc	1.24 ± 0.00 e	1.14 ± 0.02 e
Gallic acid	34.60 ± 0.42 a	4.98 ± 0.23 c	nd	nd	nd	6.02 ± 0.13 b	nd	nd

nd—no detected; Results are presented as the mean (*n* = 3) ± SD and those with different letters are significantly different, with *p* < 0.05; E1 S–extraction one step.

**Table 8 foods-11-03205-t008:** Individual phenolic compounds obtained for group 2 extractions by Two-step extraction–E2S.

Phenolic Compounds (mg/100 g DM)	DES1	DES2	DES3	DES4	DES5	DES6	DES7
Naringin	nd	nd	2.77 ± 0.06 b	nd	nd	nd	nd
Narirutin	33.61 ± 2.19 def	21.58 ± 0.16 f	29.44 ± 1.47 ef	48.97 ± 5.90 ef	26.90 ± 2.43 ef	25.52 ± 0.18 ef	15.56 ± 0.13 f
Hesperidin	269.72 ± 5.35 d	151.02 ± 0.10 hi	nd	299.27 ± 6.76 fg	360.16 ± 9.74 c	256.24 ± 0.39 h	142.17 ± 0.91 hi
Ellagic acid	9.59 ± 0.35 a	2.24 ± 0.18 f	nd	2.12 ± 0.36 f	6.74 ± 0.27 c	8.33 ± 0.14 abc	0.31 ± 0.10 g
Naringenin	82.26 ± 0.80 a	42.92 ± 0.07 def	75.56 ± 0.35 cd	56.17 ± 1.49 bc	27.35 ± 2.50 bc	78.71 ± 0.11 b	70.04 ± 0.14 f
Hesperitin	83.55 ± 0.41 a	46.50 ± 0.20 c	76.81 ± 0.39 c	24.99 ± 1.39 d	21.65 ± 0.61 f	51.57 ± 0.01 b	23.44 ± 0.18 de
Diosmetin	6.54 ± 0.16 a	nd	2.69 ± 0.06 c	nd	2.66 ± 0.00 c	2.72 ± 0.02 c	nd
Tangeritin	nd	nd	2.08 ± 0.08 a	1.89 ± 0.00 b	nd	1.62 ± 0.07 c	nd
Gallic acid	5.82 ± 0.13 b	6.03 ± 0.22 b	3.52 ± 0.12 d	4.66 ± 0.04 c	nd	3.75 ± 0.17 d	3.86 ± 0.09 d

nd—no detected; Results are presented as the mean (*n* = 3) ± SD and those with different letters are significantly different, with *p* < 0.05; E2E–Extraction 2 steps.

**Table 9 foods-11-03205-t009:** Antioxidant activity.

DES or Extractive Mixture	FRAP (µmol TE/g DM)		DPPH (µmol TE/g DM)	
	E1S	E2S	E1S	E2S
Water:ethanol (1:1)	6920 ± 721.69 e	-	16,682.04 ± 2139 a	-
DES 1	15,981 ± 877.57 b	23,200 ± 721.69 a	7647 ± 237.47 bcde	10,545 ± 75.47 b
DES 2	3061.50 ± 184.46 g	7566.70 ± 649.52 e	6076 ± 1423.41 cde	8700 ± 1977.86 bcde
DES 3	987.50 ± 190.94 h	3132.80 ± 272.09 g	5058.52 ± 1641.07 e	6956 ± 1277.05 bcde
DES 4	7370.00 ± 288.68 e	10,725 ± 741.23 d	6851 ± 781.31 bcde	9776. ± 822.30 bc
DES 5	12,890 ± 697.87 c	15,302.10 ± 721.25 b	7547 ± 642.30 bcde	10,138.10 ± 593.35 b
DES 6	5058.30 ± 288.68 f	7836.70 ± 288.68 e	6110.73 ± 479.09 cde	9587.15 ± 393.51 bcd
DES 7	2496 ± 232.84 gh	7210 ± 288.68 e	5901 ± 931.87 de	8083.37 ± 2169 bcde

Tukey test performed between all values of the two extractions, separately for all DPPH results and then for all FRAP results. Results are presented as the mean (*n* = 3) ± SD and those with different letters are significantly different, with *p* < 0.05.

## Data Availability

Data is contained within the article.

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
