# Peer review of "High Concentrate Flavonoids Extract from Citrus Pomace Using Enzymatic and Deep Eutectic Solvents Extraction"

_foods, 2022, doi:10.3390/foods11203205_

Round 1

Reviewer 1 Report

The manuscript entitled “High Concentrate Flavonone Extract from Citrus Pomace Using Enzymatic and Deep Eutectic Solvents Extraction” is interesting for the phenolic compound extraction. However, it does not seem that there is a logic in these analysis of the results about the group 1, group 2 and comparison group. The presentations of some results are confusing. More importantly, the results were described in detail, however, most of the results were lack of in-depth discussion. Hence, I don’t think it could be published this actual format. The detailed comments are as follows:

 1.    It confuses CPWP with CPNP in this manuscript, such as section 2.1, table 2. Please check the whole manuscript.

2.    There is no line number in the manuscript, which makes it inconvenient to make comments.

3.    Key results should be summarized in the abstract.

4.    The carbohydrate content in CPNP is higher than the carbohydrate content in CPWP, why?

5.    Section 3.2.1, Table 3 shows the results of the total phenolic compound…….. pomace with and without pectin, respectively. Where is the data for total phenolic compound of pomace without pectin in table 3?

6.    In table 3, the labelling letter of the significant difference in the second column needs to be checked.

7.    Figures in Figure 2 are too small and hardly read. It might be suitable for using table.

8.    The HPLC chromatogram should be giving.

9.    Figures in table 6 are too messy and hardly read.

10.  There are many format errors in the manuscript, such as table 4.

Reviewer 2 Report

General comments: this article studies the extraction of phenolic compounds by applying DES (Deep Eutectic Solvents) combined with enzymatic treatment from citrus pomace. Objectives in the introduction are not well established. Materials and methods are well organized, needing minor revisions. Results need major revision. Conclusions are, mainly, a summary of the results.

My English is no perfect, but it can be observed that this article needs English revision.

I recommend some changes to improve it:

Title: All of the compounds extracted belongs to flavonones? If not, flavonone should be replace by other word.

Abstract:

The extraction procedure is too detailed in this section, being more important the comparison with conventional extraction, that is not even mentioned here.

Introduction:

Two paragraphs mentioning Yazici and Osmen, and Qi et al., should be better part of discussion, not introduction. The authors can even delete these two paragraphs.

Phenolic compounds groups extracted are not mentioned in the introduction. Are all of them belonging to flavonoids? If so, the abstract should mention them and not phenolic compounds in general.

The reason to compare the extractions techniques (conventional solvent against DES and enzyme treatment) should be better justified.

The interest of the 3 main extracted phenolic compounds should be mentioned in the introduction.

Materials and methods:

2.1 Materials and samples should be replaced by 2.1 Reagents and samples

Reagents for DPPH and Frap should be included in this subsection. Also, ethanol and formic acid must be mentioned here.

Why chemical composition was determined? It seems that the influence of chemical composition on the polyphenol compounds extraction processes is not relevant.

In 2.4.1 subsection, “with filter of 725 nm” is not understandable.

Were the analyses performed in duplicate, triplicate or both? Please specify. For example, in subsection 2.4.1 n=3 and in section 2.5 the authors do not specify the number of replicates.

In subsection 2.4.2, replace “allowed detection at 280 nm” by “allowed detection from ?? nm to ?? nm”.

In subsection 2.4.3 please write Trolox equivalents in the same way for DPPH and for FRAP.

Results:

In subsection 3.3.1, please replace “Novozym” by enzyme.

Author Response

"Please see the attachment

Round 2

Reviewer 1 Report

The authors made several improvements after revision, still some issues are remaining. For example, the authors did not manage to make significant improvements in the manuscript following author’s guideline of MDPI. So the manuscript needed further optimization and improved. Some comments are as follows:

 1.    Line numbers should be continuing in the text.

2.     Conclusion in the abstract should be included. it Indicates the main conclusions or interpretations of the study.

3.     Reference numbers should be placed before the punctuation, please check throughout the text.  

4.     In page 3 line 10, “2.2. Chemical composition of pomace” should be change to “2.4.1 Determination of chemical composition of pomace”.

Reviewer 2 Report

The paper is now improved. I only recomend 2 changes: the comparison with conventional extraction is not mentioned in the abstract neither in the introduction. In page 7, the sentence in line 2 is wrongly written.
